# UNIVERSAL APPROXIMATION THEOREM FOR EQUIVARIANT MAPS BY GROUP CNNS

## ABSTRACT

Group symmetry is inherent in a wide variety of data distributions. Data processing that preserves symmetry is described as an equivariant map and often effective in achieving high performance. Convolutional neural networks (CNNs) have been known as models with equivariance and shown to approximate equivariant maps for some specific groups. However, universal approximation theorems for CNNs have been separately derived with individual techniques according to each group and setting. This paper provides a unified method to obtain universal approximation theorems for equivariant maps by CNNs in various settings. As its significant advantage, we can handle non-linear equivariant maps between infinite-dimensional spaces for non-compact groups.

## 1 INTRODUCTION

Deep neural networks have been widely used as models to approximate underlying functions in various machine learning tasks. The expressive power of fully-connected deep neural networks was first mathematically guaranteed by the universal approximation theorem in Cybenko (1989), which states that any continuous function on a compact domain can be approximated with any precision by an appropriate neural network with sufficient width and depth. Beyond the classical result stated above, several types of variants of the universal approximation theorem have also been investigated under different conditions.

Among a wide variety of deep neural networks, convolutional neural networks (CNNs) have achieved impressive performance for real applications. In particular, almost all of state-of-the-art models for image recognition are based on CNNs. These successes are closely related to the property that performing CNNs commute with translation on pixel coordinate. That is, CNNs can conserve symmetry about translation in image data. In general, this kind of property for symmetry is known as the *equivariance*, which is a generalization of the *invariance*. When a data distribution has some symmetry and the task to be solved relates to the symmetry, data processing is desired to be equivariant on the symmetry. In recent years, different types of symmetry have been focused per each task, and it has been proven that CNNs can approximate arbitrary equivariant data processing for specific symmetry. These results are mathematically captured as the universal approximation for equivariant maps and represent the theoretical validity of the use of CNNs.

In order to theoretically correctly handle symmetric structures, we have to carefully consider the structure of data space where data distributions are defined. For example, in image recognition tasks, image data are often supposed to have symmetry for translation. When each image data is acquired, there are finite pixels equipped with an image sensor, and an image data is represented by a finite-dimensional vector in a Euclidean space $\mathbb{R}^d$, where $d$ is the number of pixels. However, we note that the finiteness of pixels stems from the limit of the image sensor and a raw scene behind the image data is thought to be modelled by an element in $\mathbb{R}^\mathcal{S}$ with continuous spatial coordinates $\mathcal{S}$, where $\mathbb{R}^\mathcal{S}$ is a set of functions from $\mathcal{S}$ to $\mathbb{R}$. Then, the element in $\mathbb{R}^\mathcal{S}$ is regarded as a functional representation of the image data in $\mathbb{R}^d$. In this paper, in order to appropriately formulate data symmetry, we treat both typical data representation in finite-dimensional settings and functional representation in infinite-dimensional settings in a unified manner.

## 1.1 RELATED WORKS

**Symmetry and functional representation.** Symmetry is mathematically described in terms of groups and has become an essential concept in machine learning. Gordon et al. (2019) point out that, when data symmetry is represented by a infinite group like the translation group, equivariant maps, which are symmetry-preserving processing, cannot be captured as maps between finite-dimensional spaces but can be described by maps between infinite-dimensional function spaces. As a related study about symmetry-preserving processing, Finzi et al. (2020) propose group convolution of functional representations and investigate practical computational methods such as discretization and localization.

**Universal approximation for continuous maps.** The universal approximation theorem, which is the main objective of this paper, is one of the most classical mathematical theorems of neural networks. The universal approximation theorem states that a feedforward fully-connected network (FNN) with a single hidden layer containing finite neurons can approximate a continuous function on a compact subset of $\mathbb{R}^d$. Cybenko (1989) proved this theorem for the sigmoid activation function. After his work, some researchers showed similar results to generalize the sigmoidal function to a larger class of activation functions as Barron (1994), Hornik et al. (1989), Funahashi (1989), Kůrková (1992) and Sonoda & Murata (2017). These results were approximations to functional representations between finite-dimensional vector spaces, but recently Guss & Salakhutdinov (2019) generalized them to continuous maps between infinite-dimensional function spaces in Guss & Salakhutdinov (2019).

**Equivariant neural networks.** The concept of group-invariant neural networks was first introduced in Shawe-Taylor (1989) in the case of permutation groups. In addition to the invariant case, Zaheer et al. (2017a) designed group-equivariant neural networks for permutation groups and obtained excellent results in many applications. Maron et al. (2019a; 2020) consider and develop a theory of equivariant tensor networks for general finite groups. Petersen & Voigtlaender (2020) established a connection between group CNNs, which are equivariant networks, and FNNs for group finites. However, symmetry are not limited to finite groups. Convolutional neural networks (CNNs) was designed to be equivariant for translation groups and achieved impressive performance in a wide variety of tasks. Gens & Domingos (2014) proposed architectures that are based on CNNs and invariant to more general groups including affine groups. Motivated by CNN's experimental success, many researchers have further generalized this by using group theory. Kondor & Trivedi (2018) proved that, when a group is compact and the group action is transitive, a neural network constrained by some homogeneous structure is equivariant if and only if it becomes a group CNN.

**Universal approximation for equivariant maps.** Compared to the vast studies about universal approximation for continuous maps, there are few existing studies about universal approximation for equivariant maps. Sannai et al. (2019); Ravanbakhsh (2020); Keriven & Peyré (2019) considered the equivariant model for finite groups and proved universal approximation property of them by attributing it to the results of Maron et al. (2019b). Cohen et al. (2019) considered group convolution on a homogeneous space and proved that a linear equivariant map is always convolution-like. Yarotsky (2018) proved universal approximation theorems for nonlinear equivariant maps by CNN-like models when groups are the $d$-dimensional translation group $T(d) = \mathbb{R}^d$ or the 2-dimensional Euclidean group SE(2). However, when groups are more general, universal approximation theorems for non-linear equivariant maps have not been obtained.

## 1.2 PAPER ORGANIZATION AND OUR CONTRIBUTIONS

The paper is organized as follows. In section 2, we introduce the definition of group equivariant maps and provide the essential property that equivariant maps have one-to-one correspondence to theoretically tractable maps called generators. In section 3, we define fully-connected and group convolutional neural networks between function spaces. This formulation is suitable to represent data symmetry. Then, we provide a main theorem called the conversion theorem that can convert FNNs to CNNs. In section 4, using the conversion theorem, we derive universal approximation theorems for non-linear equivariant maps by group CNNs. In particular, this is the first universal approximation theorem for equivariant maps in infinite-dimensional settings. We note that finite and infinite groups are handled in a unified manner. In section 5, we provide concluding remarks and mention future works.

## 2 GROUP EQUIVARIANCE

### 2.1 PRELIMINARIES

We introduce definitions and terminology used in the later discussion.

**Functional representation.** In this paper, sets denoted by $\mathcal{S}$, $\mathcal{T}$ and $G$ are assumed to be locally compact, $\sigma$-compact, Hausdorff spaces. When $\mathcal{S}$ is a set, we denote by $\mathbb{R}^{\mathcal{S}}$ the set of all maps from $\mathcal{S}$ to $\mathbb{R}$ and by $\| \cdot \|_{\infty}$ the supremum norm. We call $\mathcal{S}$ of $\mathbb{R}^{\mathcal{S}}$ the index set. We denote by $\mathcal{C}(\mathcal{S})$ the set of all continuous maps from $\mathcal{S}$ to $\mathbb{R}$. We denote by $\mathcal{C}_0(\mathcal{S})$ the set of continuous functions from $\mathcal{S}$ to $\mathbb{R}$ which vanish at infinity[1]. For a Borel space $\mathcal{S}$ with some measure $\mu$, we denote the set of integrable functions from $\mathcal{S}$ to $\mathbb{R}$ with respect to $\mu$ as $L^1_{\mu}(\mathcal{S})$. For a subset $\mathcal{B} \subset \mathcal{S}$, the restriction map $R_{\mathcal{B}} : \mathbb{R}^{\mathcal{S}} \to \mathbb{R}^{\mathcal{B}}$ is defined by $R_{\mathcal{B}}(x) = x|_{\mathcal{B}}$, where $x \in \mathbb{R}^{\mathcal{S}}$ and $x|_{\mathcal{B}}$ is the restriction of the domain of $x$ onto $\mathcal{B}$.

When $\mathcal{S}$ is a finite set, $\mathbb{R}^{\mathcal{S}}$ is identified with the finite-dimensional Euclidean space $\mathbb{R}^{|\mathcal{S}|}$, where $|\mathcal{S}|$ is the cardinality of $\mathcal{S}$. In this sense, $\mathbb{R}^{\mathcal{S}}$ for general sets $\mathcal{S}$ is a generalization of Euclidean spaces. However, $\mathbb{R}^{\mathcal{S}}$ itself is often intractable for an infinite set $\mathcal{S}$. In such cases, we instead consider $\mathcal{C}(\mathcal{S})$, $\mathcal{C}_0(\mathcal{S})$ or $L^p(\mathcal{S})$ as relatively tractable subspaces of $\mathbb{R}^{\mathcal{S}}$.

**Group action.** We denote the identity element in a group $G$ by $1$. We assume that the action of a group $G$ on a set $\mathcal{S}$ is continuous. We denote by $g \cdot s$ the left action of $g \in G$ to $s \in \mathcal{S}$. Then we call $G_s := \{g \cdot s | g \in G\}$ the orbit of $s \in \mathcal{S}$. From the definition, we have $\mathcal{S} = \bigcup_{s \in \mathcal{S}} G_s$. When a subset $\mathcal{B} \subset \mathcal{S}$ is the set of representative elements from all orbits, it satisfies the disjoint condition $\mathcal{S} = \bigsqcup_{s \in \mathcal{B}} G_s$. Then, we call $\mathcal{B}$ a base space[2] and define the projection $P_{\mathcal{B}} : \mathcal{S} \to \mathcal{B}$ by mapping $s \in \mathcal{S}$ to the representative element in $\mathcal{B} \cap G_s$. When a group $G$ acts on sets $\mathcal{S}$ and $\mathcal{T}$, the action of $G$ on the product space $\mathcal{S} \times \mathcal{T}$ is defined by $g \cdot (s,t) := (g \cdot s, g \cdot t)$. When a group $G$ acts on a index set $\mathcal{S}$, the $G$-translation operators $T_g : \mathbb{R}^{\mathcal{S}} \to \mathbb{R}^{\mathcal{S}}$ for $g \in G$ are defined by $T_g[x](s) := x(g^{-1} \cdot s)$, where $x \in \mathbb{R}^{\mathcal{S}}$ and $s \in \mathcal{S}$. We often denote $T_g[x]$ simply by $g \cdot x$ for brevity. Then, group translation determine the action[3] of $G$ on $\mathbb{R}^{\mathcal{S}}$.

### 2.2 GROUP EQUIVARIANT MAPS

In this section, we introduce group equivariant maps and show their basic properties. First, we define group equivariance.

**Definition 1** (Group Equivariance). *Suppose that a group $G$ acts on sets $\mathcal{S}$ and $\mathcal{T}$. Then, a map $F : \mathbb{R}^{\mathcal{S}} \to \mathbb{R}^{\mathcal{T}}$ is called $G$-equivariant when $F[g \cdot x] = g \cdot F[x]$ holds for any $g \in G$ and $x \in \mathbb{R}^{\mathcal{S}}$.*

An example of an equivariant map in image processing is provided in Figure 1.

To clarify the degree of freedom of equivariant maps, we define the generator of equivariant maps.

**Definition 2** (Generator). *Let $\mathcal{B} \subset \mathcal{T}$ be a base space with respect to the action of $G$ on $\mathcal{T}$. For a $G$-equivariant map $F : \mathbb{R}^{\mathcal{S}} \to \mathbb{R}^{\mathcal{T}}$, we call $F_{\mathcal{B}} := R_{\mathcal{B}} \circ F$ the generator of $F$.*

The following theorem shows that equivariant maps can be represented by their generators.

**Theorem 3** (Degree of Freedom of Equivariant Maps). *Let a group $G$ act on sets $\mathcal{S}$ and $\mathcal{T}$, and $\mathcal{B} \subset \mathcal{T}$ a base space. Then, a $G$-equivariant map $F : \mathbb{R}^{\mathcal{S}} \to \mathbb{R}^{\mathcal{T}}$ has one-to-one correspondence to its generator $F_{\mathcal{B}}$.*

A detailed version of Theorem 3 is proved in Section A.1.

---

[1]A function $f$ on a locally compact space is said to vanish at infinity if, for any $\epsilon$, there exists a compact subset $\mathcal{K} \subset \mathcal{S}$ such that $\sup_{s \in \mathcal{S} \setminus \mathcal{K}} |f(s)| < \epsilon$.

[2]The choice of the base space is not unique in general. However, the topological structure of a base space can be induced by the quotient space $\mathcal{S}/G$.

[3]We note that $T_g \circ T_{g'} = T_{g'g}$ and the group translation operator is the action of $G$ on $\mathbb{R}^{\mathcal{S}}$ from the right.

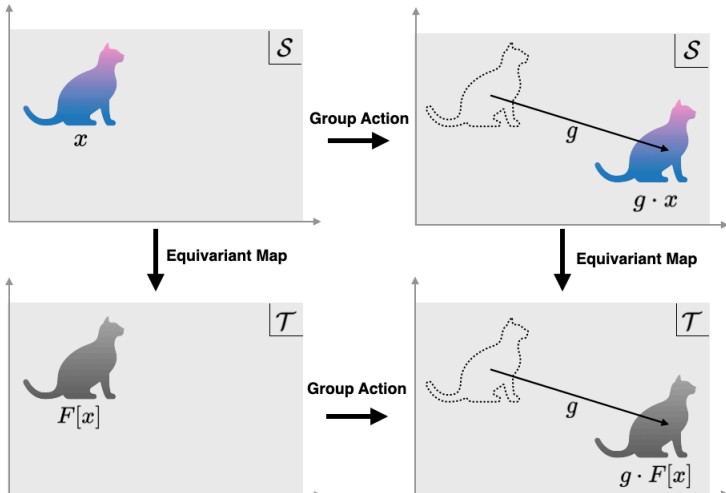

Figure 1: An example of an equivariant map from RGB images to gray-scale images. An RGB image $x$ is represented by values (i.e., a function) on 2-dimensional spatial coordinates with RGB channels. This corresponds to the case where the index set is $\mathcal{S} = \mathbb{R}^{2\times3} = \mathbb{R}^6$. Similarly, a gray-scale image $F[x]$ after equivariant processing $F : \mathbb{R}^{\mathcal{S}} \to \mathbb{R}^{\mathcal{T}}$ is represented by values on 2-dimensional spatial coordinates with a single gray-scale channel. This corresponds to the case where the index set is $\mathcal{T} = \mathbb{R}^2$. In this figure, the group action is translation of $G = \mathbb{R}^2$ to 2-dimensional spatial coordinates.

## 3 FULLY-CONNECTED AND GROUP CONVOLUTIONAL NEURAL NETWORKS

### 3.1 FULLY-CONNECTED NEURAL NETWORKS

To define neural networks, we introduce some notions. A map $A : \mathbb{R}^{\mathcal{S}} \to \mathbb{R}^{\mathcal{T}}$ is called a bounded affine map if there exist a bounded linear map $W : \mathbb{R}^{\mathcal{S}} \to \mathbb{R}^{\mathcal{T}}$ and an element $b \in \mathbb{R}^{\mathcal{T}}$ such that

$$A[x] = W[x] + b. \tag{1}$$

Guss & Salakhutdinov (2019) provide the following lemma, which is useful to handle bounded affine maps.

**Lemma 4** (Integral Form, Guss & Salakhutdinov (2019)). *Suppose that $\mathcal{S}$ and $\mathcal{T}$ are locally compact, $\sigma$-compact, Hausdorff, measurable spaces. For a bounded linear map $W : \mathcal{C}(\mathcal{S}) \to \mathcal{C}(\mathcal{T})$, there exist a Borel regular measure $\mu$ on $\mathcal{S}$ and a weak\* continuous family of functions $\{w(t, \cdot)\}_{t\in\mathcal{T}} \subset L^1_\mu(\mathcal{S})$ such that the following holds for any $x \in \mathcal{C}(\mathcal{S})$:*

$$W[x](t) = \int_{\mathcal{S}} w(t, s)x(s)d\mu(s).$$

To use the integral form, we assume in the following that the input and output spaces of $A$ are the class of continuous maps $\mathcal{C}(\mathcal{S})$ and $\mathcal{C}(\mathcal{T})$ instead of $\mathbb{R}^{\mathcal{S}}$ and $\mathbb{R}^{\mathcal{T}}$, respectively. Using the integral form, a bounded affine map $A$ is represented by

$$A_{\mu,w,b}[x](t) = \int_{\mathcal{S}} w(t, s)x(s)d\mu(s) + b(t). \tag{2}$$

In particular, when $\mathcal{S}$ and $\mathcal{T}$ are finite sets with cardinality $d$ and $d'$, the function spaces $\mathcal{C}(\mathcal{S})$ and $\mathcal{C}(\mathcal{T})$ are identified with finite-dimensional Euclidean spaces $\mathbb{R}^d$ and $\mathbb{R}^{d'}$, and thus, an affine map $A : \mathbb{R}^d \to \mathbb{R}^{d'}$ is parameterized by a weight matrix $W = [w(t, s)]_{s\in[d],t\in[d']} : \mathbb{R}^d \to \mathbb{R}^{d'}$ and a bias vector $b = [b(t)]_{t\in[d']} \in \mathbb{R}^{d'}$, and (2) induces the following form, which is often used in the literature on neural networks:

$$A[x](t) = \sum_{s=1}^{d} w(t, s)x(s) + b(t). \tag{3}$$

A continuous function $\rho : \mathbb{R} \to \mathbb{R}$ induces the activation map $\alpha_\rho : \mathcal{C}(\mathcal{S}) \to \mathcal{C}(\mathcal{S})$ which is defined by $\alpha_\rho(x) := \rho \circ x \in \mathcal{C}(\mathcal{S})$ for $x \in \mathcal{C}(\mathcal{S})$. However, for brevity, we denote $\alpha_\rho$ by $\rho$. Then, we can define fully-connected neural networks in general settings.

**Definition 5** (Fully-connected Neural Networks). *Let $L \in \mathbb{N}$. A fully-connected neural network with $L$ layers is a composition map of bounded affine maps $(A_1, \ldots, A_L)$ and an activation map $\rho$ represented by*

$$\phi := A_L \circ \rho \circ A_{L-1} \circ \cdots \circ \rho \circ A_1, \tag{4}$$

*where $A_\ell : \mathcal{C}(\mathcal{S}_{\ell-1}) \to \mathcal{C}(\mathcal{S}_\ell)$ are affine maps for some sequence of sets $\{\mathcal{S}_\ell\}_{\ell=0}^L$. Then, we denote by $\mathcal{N}_{\mathsf{FNN}}(\rho, L; \mathcal{S}_0, \mathcal{S}_L)$ the set of all fully-connected neural networks from $\mathcal{C}(\mathcal{S}_0)$ to $\mathcal{C}(\mathcal{S}_L)$ with $L$ layers and an activation function $\rho$.*

We denote the measure of the affine map $A_1$ in the first layer of a fully-connected neural network $\phi$ by $\mu_\phi$. This measure $\mu_\phi$ is used to describe a condition in the main theorem (Theorem 9).

### 3.2 GROUP CONVOLUTIONAL NEURAL NETWORKS

We introduce the general form of group convolution.

**Definition 6** (Group Convolution). *Suppose that a group $G$ acts on sets $\mathcal{S}$ and $\mathcal{T}$. For a $G$-invariant measure $\nu$ on $\mathcal{S}$, $G$-invariant functions $v : \mathcal{S} \times \mathcal{T} \to \mathbb{R}$ and $b \in \mathcal{C}(\mathcal{T})$, the biased $G$-convolution $C_{\nu,v,b} : \mathcal{C}(\mathcal{S}) \to \mathcal{C}(\mathcal{T})$ is defined as*

$$C_{\nu,v,b}[x](t) := \int_{\mathcal{S}} v(t,s) x(s) d\nu(s) + b(t). \tag{5}$$

*In the right hand side, we call the first term the $G$-convolution and the second term the bias term.*

In the following, we denote $C_{\nu,v,b}$ by $C$ for brevity. When $\mathcal{S}$ and $\mathcal{T}$ are finite, we note that (5) also can be represented as (3).

Definition 6 includes existing definitions of group convolution as follows. When $\mathcal{S} = \mathcal{T} = G$, the group $G$ acts on $\mathcal{S}$ and $\mathcal{T}$ by left translations. Then, (5) without the bias term (i.e., $b = 0$) is described as

$$C[x](g) = \int_{\mathcal{G}} v(g,h) x(h) d\nu(h) = \int_{\mathcal{G}} \tilde{v}(h^{-1}g) x(h) d\nu(h),$$

where[4] $\tilde{v}(g) := v(g,1)$. This is a popular definition of group convolution between two functions on $G$. Further, when $\mathcal{S} = G \times \mathcal{B}$ and $\mathcal{T} = G \times \mathcal{B}'$, (5) without the bias term is described as

$$C[x](g,t) = \int_{\mathcal{G} \times \mathcal{B}} v((g,\tau),(h,\varsigma)) x(h,\varsigma) d\nu(h,\varsigma) = \int_{\mathcal{G} \times \mathcal{B}} \tilde{v}(h^{-1}g, \tau, \varsigma) x(h,\varsigma) d\nu(h,\varsigma),$$

where $\tilde{v}(g,\tau,\varsigma) := v((g,\tau),(1,\varsigma))$. This coincides with the definition of group convolution in Finzi et al. (2020). We note that Finzi et al. (2020) also proposes discretization and localization of the above group convolution for implementation.

In conventional convolution used for image recognition, $G$ represents spatial information such as pixel coordinate, $\mathcal{B}$ and $\mathcal{B}'$ correspond to channels in consecutive layers $\ell$ and $\ell + 1$ respectively, and $v$ corresponds to a filter. In applications, the filter $v$ is expected to have compact support or be short-tailed on $G$ as in a $3 \times 3$ convolution filter in discrete convolution. In particular, when $v$ is allowed to be the Dirac delta or highly peaked around a single point in $G$, such convolution can be interpreted as the $1 \times 1$ convolution.

Then, we define group convolutional neural networks as follows.

**Definition 7** (Group Convolutional Neural Networks). *Let $L \in \mathbb{N}$. A $G$-convolutional neural network with $L$ layers is a composition map of biased convolutions $C_\ell : \mathcal{C}(\mathcal{S}_{\ell-1}) \to \mathcal{C}(\mathcal{S}_\ell)$ ($\ell = 1, \ldots, L$) for some sequence of spaces $\{\mathcal{B}_\ell\}_{\ell=0}^L$ and an activation map with $\rho$ as*

$$\Phi := C_L \circ \rho \circ C_{L-1} \circ \cdots \circ \rho \circ C_1. \tag{6}$$

*Then, we denote by $\mathcal{N}_{\mathsf{CNN}}(G, \rho, L; \mathcal{S}_0, \mathcal{S}_L)$ the set of all $G$-convolutional neural networks from $\mathcal{C}(\mathcal{S}_0)$ to $\mathcal{C}(\mathcal{S}_L)$ with respect to a group $G$ with $L$ layers and a fixed activation function $\rho$.*

---

[4]A bivariate $G$-invariant function $v : G \times G \to \mathbb{R}$ is determined by the univariate function $\tilde{v} : G \to \mathbb{R}$ because $v(g,h) = v(h^{-1}g, h^{-1}h) = v(h^{-1}g, 1) = \tilde{v}(h^{-1}g)$.

We easily verify the following proposition.

**Proposition 8.** *A $G$-convolutional neural network is $G$-equivariant.*

In particular, each biased $G$-convolution $C_{\nu,v,b}$ is $G$-equivariant. Conversely, Cohen et al. (2019) showed that a $G$-equivariant *linear* map is represented by some $G$-convolution without the bias term when $G$ is locally compact and unimodular, and the action of a group is transitive (i.e., $\mathcal{B}$ consists of only a single element).

### 3.3  CONVERSION THEOREM

In this section, we introduce the main theorem (Theorem 9), which is an essential part of obtaining universal approximation theorems for equivariant maps by group CNNs.

**Theorem 9** (Conversion Theorem). *Suppose that the action of a group $G$ on sets $\mathcal{S}$ and $\mathcal{T}$. We assume the following condition:*

> *(C1)  there exist base spaces $\mathcal{B}_{\mathcal{S}} \subset \mathcal{S}$, $\mathcal{B}_{\mathcal{T}} \subset \mathcal{T}$, and two subgroups[5] $H_{\mathcal{T}} \leqslant H_{\mathcal{S}} \leqslant G$ such that $\mathcal{S} = G/H_{\mathcal{S}} \times \mathcal{B}_{\mathcal{S}}$ and $\mathcal{T} = G/H_{\mathcal{T}} \times \mathcal{B}_{\mathcal{T}}$.*

*Further, suppose $E \subset \mathcal{C}_0(\mathcal{S})$ is compact and an FNN $\phi : E \to \mathcal{C}_0(\mathcal{B}_{\mathcal{T}})$ with a Lipschitz activation function $\rho$ satisfies*

> *(C2)  there exists a $G$-left-invariant locally finite measure $\nu$ on $\mathcal{S}$ such that[6] $\mu_\phi \ll \nu$.*

*Then, for any $\epsilon > 0$, there exists a CNN $\Phi : E \to \mathcal{C}_0(\mathcal{T})$ with the activation function $\rho$ such that the number of layers of $\Phi$ equals that of $\phi$ and*

$$\|R_{\mathcal{B}_{\mathcal{T}}} \circ \Phi - \phi\|_\infty \leq \epsilon. \tag{7}$$

*Moreover, for any $G$-equivariant map $F : \mathcal{C}_0(\mathcal{S}) \to \mathcal{C}_0(\mathcal{T})$, the following holds:*

$$\|F|_E - \Phi\|_\infty \leq \|F_{\mathcal{B}_{\mathcal{T}}}|_E - \phi\|_\infty + \epsilon. \tag{8}$$

We provide the proof of Theorem 9 in Section B.

**Conversion of Universal Approximation Theorems**. The conversion theorem can convert a universal approximation theorem by FNNs to a universal approximation theorem for equivariant maps by CNNs as follows. Suppose that the existence of an FNN $\phi$ which satisfies $\|F_{\mathcal{B}}|_E - \phi\|_\infty \leq \epsilon$ using some universal approximation theorem by FNNs. Then, Theorem 9 guarantees the existence of a CNN $\Phi$ which satisfies $\|F|_E - \Phi\|_\infty \leq 2\epsilon$. In other words, if an FNN can approximate the generator of the target equivariant map on $E$, then there exists a CNN which approximates the whole of the equivariant map on $E$.

**Applicable Cases**. The conversion theorem can be applied to a wide range of group actions. We explain the generality of the conversion theorem. First, sets $\mathcal{S}$ and $\mathcal{T}$ are not limited to finite sets or Euclidean spaces, and may be more general topological spaces. Second, a group $G$ may be discrete (especially finite) or continuous groups. Moreover, $G$ can be non-compact and non-commutative. Third, the action of a group $G$ on sets $\mathcal{S}$ and $\mathcal{T}$ may not be transitive, and thus, the sets can be non-homogeneous spaces. In the following, we provide some concrete examples of group actions when $\mathcal{S} = \mathcal{T}$ and the actions of $G$ on $\mathcal{S}$ and $\mathcal{T}$ are the same:

- **Symmetric Group**. The action of $G = S_n$ on $\mathcal{S} = [n]$ as permutation has the decomposition $[n] = S_n/\mathrm{Stab}(1) \times \{*\}$, where $H_{\mathcal{S}} = \mathrm{Stab}(1)$ is the set of all permutations on $[n]$ that fix $1 \in [n]$ and $\mathcal{B}_{\mathcal{S}} = \{*\}$ is a singleton[7]. Then, the counting measure can be taken as an invariant measure $\nu$.

- **Rotation Group**. The action of $G = \mathrm{O}(d)$ on $\mathcal{S} = \mathbb{R}^d \setminus \{0\}$ as rotation around $0 \in \mathbb{R}^d$ has the decomposition $\mathbb{R}^d \setminus \{0\} = \mathrm{O}(d)/\mathrm{O}(d-1) \times \mathbb{R}_+$ The cases where $G = \mathrm{SO}(d)$ or $\mathcal{S} = S^{d-1}$ have similar decomposition. Then, the Lebesgue measure can be taken as an invariant measure $\nu$.

---

[5] $H_{\mathcal{S}}$ and $H_{\mathcal{T}}$ are not assumed to be normal subgroups.

[6] $\mu_\phi \ll \nu$ means that $\mu_\phi$ is absolutely continuous with respect to $\nu$.

[7] A singleton is a set with exactly one element.

- **Translation Group**. The action of $G = \mathbb{R}^d$ on $\mathcal{S} = \mathbb{R}^d$ as translation has the trivial decomposition $\mathbb{R}^d = \mathbb{R}^d/\{0\} \times \{*\}$. Then, the Lebesgue measure can be taken as an invariant measure $\nu$.

- **Euclidean Group**. The action of $G = \mathrm{E}(d)$ on $\mathcal{S} = \mathbb{R}^d$ as isometry has the decomposition $\mathbb{R}^d = \mathrm{E}(d)/\mathrm{O}(d) \times \{*\}$. The case where $G = \mathrm{SE}(d)$ has a similar decomposition. Then, the Lebesgue measure can be taken as an invariant measure $\nu$.

- **Scaling Group**. The action of $G = \mathbb{R}_{>0}$ on $\mathcal{S} = \mathbb{R}^d \setminus \{0\}$ as scalar multiplication has the decomposition $\mathbb{R}^d \setminus \{0\} = \mathbb{R}_{>0}/\{1\} \times S^{d-1}$. Then, the measure $\nu_{\mathrm{r}} \times \nu_{S^{d-1}}$ can be taken as an invariant measure $\nu$, where the measure $\nu_{\mathrm{r}}$ on $\mathbb{R}_{>0}$ is determined by $\nu_{\mathrm{r}}([a,b]) := \log \frac{b}{a}$ and $\nu_{S^{d-1}}$ is a uniform measure on $S^{d-1}$.

- **Lorentz Group**. The action of $G = \mathrm{SO}^+(d,1)$, a subgroup of the Lorentz group $\mathrm{O}(d,1)$, on the upper half plane[8] $\mathcal{S} = \mathbb{H}^{d+1}$ as matrix multiplication has the decomposition $\mathbb{H}^{d+1} = \mathrm{SO}^+(d,1)/\mathrm{SO}(n) \times \{*\}$. Then, the $\pi_\#(\nu^+)$ can be taken as a left-invariant measure $\nu$, where $\nu^+$ is a left-invariant measure on $\mathrm{SO}^+(d,1)$, $\pi : \mathrm{SO}^+(d,1) \to \mathrm{SO}^+(d,1)/\mathrm{SO}(d)$ is a canonical projection, and $\pi_\#(\nu^+)$ is the pushforward measure.

**Inapplicable Cases**. We explain some cases where the conversion theorem cannot be applied. First, similar to the above discussion, we consider the setting where $\mathcal{S} = \mathcal{T}$ and the actions of $G$ on $\mathcal{S}$ and $\mathcal{T}$ are the same. We note that, even if actions of $G_1$ and $G_2$ on $\mathcal{S}$ satisfy the conditions in the conversion theorem, a common invariant measure for both $G_1$ and $G_2$ may not exist. Then, a group $G$ including $G_1$ and $G_2$ as subgroups does not satisfies (C2). For example, there does not exist a common invariant measure about the actions of translation and scaling on a Euclidean space. In particular, the action of the general linear group $\mathrm{GL}(d)$ on the Euclidean space does not have locally-finite left-invariant measure on $\mathbb{R}^d$. Thus, the conversion theorem cannot applied to the case. Next, as we saw above, our model can handle convolutions on permutation groups, but not on general finite groups. This depends on whether $[n]$ can be represented by a quotient of $G$, as we will see later. This is also the case for tensor expressions of permutations, which require a different formulation.

Lastly, we consider the case where the actions of $G$ on $\mathcal{S}$ and $\mathcal{T}$ differ. Here, $\mathcal{S}$ and $\mathcal{T}$ may and may not be equal. As a representative case, we consider the invariant case. When the stabilizer in $\mathcal{T}$ satisfies $H_\mathcal{T} = G$, a $G$-equivariant map $F : \mathcal{C}_0(\mathcal{S}) \to \mathcal{C}_0(\mathcal{T})$ is said to be $G$-invariant. However, because of the condition $H_\mathcal{T} \leqslant H_\mathcal{S}$ in (C1), the conversion theorem cannot apply to the invariant case as long as $H_\mathcal{S} \neq G$. This kind of restriction is similar to existing studies, where the invariant case is separately handled from the equivariant case (Keriven & Peyré (2019); Maehara & NT (2019); Sannai et al. (2019)). In fact, we can show that the inequality (7) never hold for non-trivial invariant cases (i.e., $H_\mathcal{S} \neq G$ and $H_\mathcal{T} = G$) as follows: From $H_\mathcal{T} = G$, we have $\mathcal{B}_\mathcal{T} = \mathcal{T}$ and $R_{\mathcal{B}_\mathcal{T}} = \mathrm{id}$, and thus, (7) reduces to $\|\Phi - \phi\|_\infty \leq \epsilon$. Here, we note that $\phi$ is an FNN, which is not invariant in general, and $\Phi$ is a CNN, which is invariant. Thus, $\Phi$ cannot approximate non-invariant $\phi$ within a small error $\epsilon$. This implies that (7) does not hold for small $\epsilon$. However, whether (8) holds for the invariant case is an open problem.

**Remarks on Conditions (C1) and (C2)**. We consider the conditions (C1) and (C2).

In (C1), the subgroup $H_\mathcal{S} \leqslant G$ (resp. $H_\mathcal{T}$) represents the stabilizer group of the action of $G$ on $\mathcal{S}$ (resp. $\mathcal{T}$). Thus, (C1) requires that the stabilizer group on every point in $\mathcal{S}$ (resp. $\mathcal{T}$) is isomorphic to the common subgroup $H_\mathcal{S}$ (resp. $H_\mathcal{T}$). When the group action satisfies some moderate conditions, such a requirement is known to be satisfied for *most* points in the set. As a theoretical result, the principal orbit type theorem (cf. Theorem 1.32, Meinrenken (2003)) guarantees that, if the group action on a manifold $\mathcal{S}$ is proper and $\mathcal{S}/G$ is connected, there exist a dense subset $\mathcal{S}' \subset \mathcal{S}$ and a subgroup $H_\mathcal{S} \subset G$ called a principal stabilizer such that the stabilizer group on every point in $\mathcal{S}'$ is isomorphic to $H_\mathcal{S}$.

Further, (C1) assumes that the sets $\mathcal{S}$ and $\mathcal{T}$ have the direct product form of some coset $G/H$ and a base space $\mathcal{B}$. Then, the case where the base space $\mathcal{B}$ consists of a single point is equivalent to the condition that the set is homogeneous. In this sense, (C1) can be regarded as a relaxation of the homogeneous condition. In many practical cases, a set $\mathcal{S}$ on which $G$ acts can be regarded as such a direct product form. For example, when the action is transitive, the direct product decomposition

---

[8]The upper half plane is defined by $\mathbb{H}^{d+1} := \{(x_1, \ldots, x_{d+1}) \in \mathbb{R}^{d+1} | x_{d+1} > 0\}$.

trivially holds with the base space that consists of a single point. Even when the set $\mathcal{S}$ itself is not rigorously represented by the direct product form, removing some "small" subset $\mathcal{N} \subset \mathcal{S}$, the complement $\mathcal{S} \setminus \mathcal{N}$ can be often represented by the direct form. For example, when $G = \mathrm{O}(d)$ acts on the set $\mathcal{S} = \mathbb{R}^d$ as rotation around the origin $\mathcal{N} = \{0\}$, $\mathcal{S} \setminus \mathcal{N}$ has a direct product form as mentioned above. In applications, removing only the small subset $\mathcal{N}$ is expected to be negligible.

Next, we provide some remarks on the condition (C2). Let us consider two representative settings of a set $\mathcal{S}$. The first case is the setting where $\mathcal{S}$ is finite. When a $G$-invariant measure $\nu$ has a positive value on every singleton in $\mathcal{S}$, $\nu$ satisfies (C2) for an arbitrary measure $\mu_\phi$ on $\mathcal{S}$. In particular, the counting measure on $\mathcal{S}$ is invariant and satisfies (C2). The second case is the setting where $\mathcal{S}$ is a Euclidean space $\mathbb{R}^d$, and $\mu_\phi$ is the Lebesgue measure. Then, (C2) is satisfied with invariant measures on the Euclidean space for various group actions, including translation, rotation, scaling, and an Euclidean group.

Here, we give a general method to construct $\nu$ in (C2) for a compact-group action. When $\mu_\phi$ is locally finite and continuous[9] with respect to the action of a compact group $G$, the measure $\nu := \nu_G * \mu_\phi$ on $\mathcal{S}$ for a Haar measure $\nu_G$ on $G$ satisfies (C2), where $(\nu_G * \mu_\phi)(A) := \int_G \mu_\phi(g^{-1} \cdot A) d\nu_G(g)$.

# 4 Universal Approximation Theorems for Equivariant Maps

## 4.1 Universal Approximation Theorem in Finite Dimension

We review the universal approximation theorem in finite-dimensional settings. Cybenko (1989) derived the following seminal universal approximation theorem in finite-dimensional settings.

**Theorem 10** (Universal Approximation for Continuous Maps by FNNs, Cybenko (1989)). *Let an activation function $\rho : \mathbb{R} \to \mathbb{R}$ be non-constant, bounded and continuous. Let $F : \mathbb{R}^d \to \mathbb{R}^{d'}$ be a continuous map. Then, for any compact $E \subset \mathbb{R}^d$ and $\epsilon > 0$, there exists a two-layer fully connected neural network $\phi_E \in \mathcal{N}_{\mathsf{FNN}}(\rho, 2; [d], [d'])$ such that $\|F|_E - \phi_E\|_\infty < \epsilon$.*

Since $\mathcal{C}_0(\mathcal{S}) = \mathbb{R}^{|\mathcal{S}|}$ for a finite set $\mathcal{S}$, we obtain the following theorem by combining Theorem 9 with Theorem 10.

**Theorem 11** (Universal Approximation for Equivariant Continuous Maps by CNNs). *Let an activation function $\rho : \mathbb{R} \to \mathbb{R}$ be non-constant, bounded and Lipschitz continuous. Suppose that a finite group $G$ acts on finite sets $\mathcal{S}$ and $\mathcal{T}$ and (C1) in Thoerem 9 holds. Let $F : \mathbb{R}^{|\mathcal{S}|} \to \mathbb{R}^{|\mathcal{T}|}$ be a $G$-equivariant continuous map. For any compact set $E \subset \mathbb{R}^{|\mathcal{S}|}$ and $\epsilon > 0$, there exists a two-layer convolutional neural network $\Phi_E \in \mathcal{N}_{\mathsf{CNN}}(\rho, 2; |\mathcal{S}|, |\mathcal{T}|)$ such that $\|F|_E - \Phi_E\|_\infty < \epsilon$.*

We note that Petersen & Voigtlaender (2020) obtained a similar result to Theorem 11 in the case of finite groups.

**Universality of DeepSets**. DeepSets is known as invariant/equivariant models with sets as input and is known to have universality for invariant/equivariant functions on set permutation (Zaheer et al. (2017b); Ravanbakhsh (2020)). The equiariant model is a stack of affine transformations with $W = \lambda E + \gamma \mathbf{1}$ ($\mathbf{1}$ is the all-one matrix) and bias $b = c \cdot (1, ..., 1)^\top$ and then an activation function acted on. Here, we prove the universality of DeepSets as a corollary of Theorem 11. Firstly, we consider the equivariant model of DeepSets as the one we are dealing with by setting $\mathcal{S}, \mathcal{T}\ G, H$ and $\mathcal{B}$ as follows. We set $\mathcal{S} = \mathcal{T} = [n]$, $G = S_n$, $H = \mathrm{Stab}(1) := \{s \in S_n \mid s(1) = 1\}$ and $\mathcal{B} = \{*\}$, where $\{*\}$ is a singleton. Then we can see that $\mathrm{Stab}(1)$ is a subgroup of $G$ and its left cosets $G/H = [n]$. As a set, $S_n/\mathrm{Stab}(1)$ is equal to $[n]$, and the canonical $S_n$-action on $S_n/\mathrm{Stab}(1)$ is equivalent to the permutation action on $[n]$. Therefore, $\mathcal{C}(G/H \times \mathcal{B}) = \mathcal{C}([n]) = \mathbb{R}^n$ holds, and the equivariant model of our paper is equal to that of DeepSets.

**Theorem 12.** *For any permutation equivariant function $F : \mathbb{R}^n \to \mathbb{R}^n$, a compact set $E \subset \mathbb{R}^n$ and $\epsilon > 0$, there is an equivariant model of DeepSets (or equivalently, our model) $\Phi_E : E \to \mathbb{R}^n$ such that $\|\Phi_E(x) - F|_E(x)\|_\infty < \epsilon$.*

The proof of Theorem 12 is provided in Section C.

---

[9]A measure $\mu_\phi$ is said to be continuous with respect to the action of a group $G$ if $\mu_\phi(g \cdot A)$ is continuous with respect to $g \in G$ for all Borel set $A \subset \mathcal{S}$.

### 4.2 Universal Approximation Theorem in Infinite Dimension

Guss & Salakhutdinov (2019) derived a universal approximation theorem for continuous maps by FNNs in infinite-dimensional settings. However, the universal approximation theorem in Guss & Salakhutdinov (2019) assumed that the index set $\mathcal{S}$ in the input layer and $\mathcal{T}$ in the output layer are compact. Combining the conversion theorem with it, we can derive a corresponding universal approximation theorem for equivariant maps with respect to compact groups. However, the compactness condition for $\mathcal{S}$ and $\mathcal{T}$ is a crucial shortcoming to handle the action of non-compact groups such as translation or scaling. In order to overcome the above obstacle, we can show a novel universal approximation theorem for Lipschitz maps by FNNs as follows.

**Theorem 13** (Universal Approximation for Lipschitz Maps by FNNs). *Let an activation function $\rho : \mathbb{R} \to \mathbb{R}$ be continuous and non-polynomial. Let $\mathcal{S} \subset \mathbb{R}^d$ and $\mathcal{T} \subset \mathbb{R}^{d'}$ be domains. Let $F : \mathcal{C}_0(\mathcal{S}) \to \mathcal{C}_0(\mathcal{T})$ be a Lipschitz map. Then, for any compact $E \subset \mathcal{C}_0(\mathcal{S})$ and $\epsilon > 0$, there exist $N \in \mathbb{N}$ and a two-layer fully connected neural network $\phi_E = A_2 \circ \rho \circ A_1 \in \mathcal{N}_{\mathsf{FNN}}(\rho, 2; \mathcal{S}, \mathcal{T})$ such that $A_1[\cdot] = W^{(1)}[\cdot] + b^{(1)} : E \to \mathcal{C}_0([N]) = \mathbb{R}^N$, $A_2[\cdot] = W^{(2)}[\cdot] + b^{(2)} : \mathbb{R}^N \to \mathcal{C}_0(\mathcal{T})$, $\mu_{\phi_E}$ is the Lebesgue measure, and $\|F|_E - \phi_E\|_\infty < \epsilon$.*

We provide proof of Theorem 13 in the appendix. We note that $\mathcal{S} \subset \mathbb{R}^d$ and $\mathcal{T} \subset \mathbb{R}^{d'}$ in Theorem 13 are allowed to be non-compact unlike the result in Guss & Salakhutdinov (2019). Combining Theorem 9 with Theorem 13, we obtain the following theorem.

**Theorem 14** (Universal Approximation for Equivariant Lipschitz Maps by CNNs). *Let an activation function $\rho : \mathbb{R} \to \mathbb{R}$ be Lipschitz continuous and non-polynomial. Suppose that a group $G$ acts on $\mathcal{S} \subset \mathbb{R}^d$ and $\mathcal{T} \subset \mathbb{R}^{d'}$, and (C1) and (C2) in Thoerem 9 hold for the Lebesgue measure $\mu_\phi$. Let $F : \mathcal{C}_0(\mathcal{S}) \to \mathcal{C}_0(\mathcal{T})$ be a $G$-equivariant Lipschitz map. Then, for any compact set $E \subset \mathcal{C}_0(\mathcal{S})$ and $\epsilon > 0$, there exists a two-layer convolutional neural network $\Phi_E \in \mathcal{N}_{\mathsf{CNN}}(\rho, 2; \mathcal{S}, \mathcal{T})$ such that $\|F|_E - \Phi_E\|_\infty < \epsilon$.*

Lastly, we mention some universal approximation theorems for some concrete groups. When a group $G$ is an Euclidean group $\mathrm{E}(d)$ or a special Euclidean group $\mathrm{SE}(d)$, Theorem 14 shows that group CNNs are universal approximators of $G$-equivariant maps. Although Yarotsky (2018) showed that group CNNs can approximate $\mathrm{SE}(2)$-equivariant maps, our result for $d \geq 3$ was not shown in existing studies. Since Euclidean groups can be used to represent 3D motion and point cloud, Theorem 14 can provide the theoretical guarantee of 3D data processing with group CNNs. As another example, when a group $G$ is $\mathrm{SO}^+(d, 1)$, $G$ acts on the upper half plane $\mathbb{H}^{d+1}$, which is shown to be suitable for word representations in NLP (Nickel & Kiela (2017)). Since the action of $G$ preserves the distance on $\mathbb{H}^{d+1}$, group convolution with $\mathrm{SO}^+(d, 1)$ may be useful for NLP.

## 5 Conclusion

We have considered universal approximation theorems for equivariant maps by group CNNs. To prove the theorems, we showed that an equivariant map is uniquely determined by its generator. Thus, when we can take a fully-connected neural network to approximate the generator, the approximator of the equivariant map can be described as a group CNN from the conversion theorem. In this way, the universal approximation for equivariant maps by group CNNs can be obtained through the universal approximation for the generator by FNNs. We have described FNNs and group CNNs in an abstract way. In particular, we provided a novel universal approximation theorem by FNNs in the infinite dimension, where the support of the input functions is unbounded. Using this result, we obtained the universal approximation theorem for equivariant maps for non-compact groups.

We mention future work. In Theorem 14, we assumed sets $\mathcal{S}$ and $\mathcal{T}$ to be subspaces of Euclidean spaces. However, in the conversion theorem (Theorem 9), sets $\mathcal{S}$ and $\mathcal{T}$ do not need to be subspaces of Euclidean spaces and may have a more general topological structure. Thus, if there is a universal approximation theorem in non-Euclidean spaces (Courrieu (2005); Kratsios (2019)), we may be able to combine it with the conversion theorem and derive its equivariant version. Next, we note the problem of computational complexity. Although group convolution can be implemented by, e.g., discretization and localization as in Finzi et al. (2020), such implementation cannot be applied to high-dimensional groups due to high computational cost. To use group CNNs for actual machine-learning problems, it is required to construct effective architecture for practical implementation.

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
