# OpenReview forum: "Universal Approximation Theorem for Equivariant Maps by Group CNNs"
_ICLR.cc/2021/Conference — Reject_

### Official Review · AnonReviewer2 · 2020-10-23

**Rating:** 7
**Confidence:** 3

**Review:**

The paper proves a universal approximation theorem for equivariant maps by group convolutional networks in an extremely general setting. The proof applies to discrete and continuous settings, including infinite dimensional ones.

The general idea of the proof is as follows. First, it is shown that an equivariant map is determined by its generator, i.e. its values on the orbits (so far, not surprising). This means that one only has to prove universality for the generator. Universality is then proved for the generator by a fully connected network, and separately an approximation theorem of FCNs by CNNs is proved.

Proving a result of this generality is technically very challenging. Although I am not a professional mathematician, I know that there are many pitfalls when proving results like the one in this paper. For instance, the orbit space can be topologically very complicated. Having spent about a day with this paper, I think that at least the general idea of the proof is sound, and the authors have sidestepped several technical problems I could think of using well chosen technical assumptions. My knowledge of topology and analysis is not sufficient to fully vouch for the technical correctness in every detail though.

Universal approximation theorems are considered to be an important kind of result, and this paper proves a very general one for equivariant maps. Provided the other reviewers do not surface any technical concerns I would recommend acceptance.


Typos and minor issues:
- "AAmong"

- I would recommend not putting a numbered footnote at the end of an equation, as it could be read as a power. E.g. mu_phi << nu^5. Better to put the footnote after the last English word or something like that.

- On condition C1: if H_S represents the stabilizer subgroup and this group is the same at all points, then it seems to me the condition is similar to assuming that S is a homogeneous space for G, as has often been done in previous work. Would it be possible to just make that assumption?

----
Update:
I do agree with the other reviewers that the paper may be difficult to read, especially for non experts. Nevertheless I still think the paper makes a nice contribution, so I will keep my rating.

---

> ### Author Response · Authors · 2020-11-22
> **To Reviewer 2**
>
> **Comment 2-1**
> >I would recommend not putting a numbered footnote at the end of an equation, as it could be read as a power. E.g. $\mu_\phi \ll \nu^5$. Better to put the footnote after the last English word or something like that.
>
> We modified the position of the footnotes as suggested.
>
> **Comment 2-2**
> >On condition C1: if $H_S$ represents the stabilizer subgroup and this group is the same at all points, then it seems to me the condition is similar to assuming that $S$ is a homogeneous space for $G$, as has often been done in previous work. Would it be possible to just make that assumption?
>
> A set $\mathcal{S}$ is homogeneous with respect to the action of a group $G$ if and only if the number of orbits is 1 (i.e., the base space $\mathcal{B}$ consists of a single point.).
> However, in general, a group action $G$ on a set $\mathcal{S}$ have multiple orbits such as the scalar multiplication of $G=\mathbb{R}_+$ or the rotation of $G=SO(d)$ on $\mathcal{S}=\mathbb{R}^d$.
> Therefore, the condition (C1) enables us to treat non-homogeneous spaces unlike the most of previous works.
> We added the explanation about this in "Applicable Cases" of Section 3.3.

---

### Official Review · AnonReviewer1 · 2020-10-26
**Important problem, interesting results but very hard to follow**

**Rating:** 5
**Confidence:** 3

**Review:**

After discussions:
I read the author's response and other reviews. The authors made a considerable effort to address my concerns. As a result, the paper has improved and I increased my score.

Having said that, I am still not sure that the paper is ready for publication in its current form. My main concern is still the accessibility/readability of the results in this paper, which I think can be further improved for the benefit of both the community and the authors (more accessible paper => more imapct). To conclude, the results in this paper are interesting and are strong enough to warrant a publication, but the paper can really benefit from another revision.

Summary:

The paper studies the approximation power of equivariant neural networks in a very general setup: the input space is assumed to be a function space (compared to a finite-dimensional space in standard setups) and the group is assumed to be any locally compact group. The main theorem shows how to convert fully connected networks to equivariant networks. This is then used to derive universal approximation theorems from the well-known results on fully connected networks. The paper seems to suggest an interesting approach and the results seem significant but the paper is very hard to follow and I am not sure I understood it. See the comments below.

Strong points:

Important problem - invariant/equivariant models provide a very helpful inductive bias for many tasks on symmetric inputs. Understanding their approximation power is crucial.
The idea of the main theorem, i.e. converting and FNN to CNN is new and might be useful.


Weak points:

The paper is written badly and is very hard to follow. (1) the intro does not give enough information about what is actually done in the paper. The first time you see the contributions are in a short list at the end of the section. It is very hard to understand them this way. (2) Section 2.2 is unnecessarily overcomplicated and difficult to read, partially due to excessive notation. I think that the basic idea of this subsection can be written in a much simpler way. Discussing the invariant case might also help. I would also suggest rewriting the paper in the following way: start with S,T finite and present all the results. Then discuss the generalization to infinite S,T (which are less useful in my opinion).
General comment: *Please add examples* - this will help the reader follow the paper. Perhaps choosing a particular case of S,T an G and exemplifying any result in the paper on this setup can be useful.
Relation to prior work is not discussed. What is the relation of theorem 13 to the results obtained by Maron, Keriven and Ravanbakhsh? Specifically, these works show that in some cases high order tensors are needed for universal approximation by invariant neural networks - can this result be recovered from your results?

Recommendation:

The results in this paper seem strong and important, but in its current form it cannot be accepted. I recommend a significant revision of the paper to make it more readable. I would be happy to increase my score (and confidence) in this case.


Minor comments:

There are many typos in the paper.
P. 2 - replace subrepresentation with subgroup?

---

> ### Author Response · Authors · 2020-11-22
> **To Reviewer 1**
>
> **Comment 1-1**
> >(1) the intro does not give enough information about what is actually done in the paper. The first time you see the contributions are in a short list at the end of the section. It is very hard to understand them this way.
>
> We modified the introduction and added Section 1.2 in order to explain our results without the short-list style.
>
> **Comment 1-2**
> >(2) Section 2.2 is unnecessarily overcomplicated and difficult to read, partially due to excessive notation. I think that the basic idea of this subsection can be written in a much simpler way.
>
> We carefully checked Sections 2.1 and 2.2
> and modified as follows:
> 1. The indicator function defined in Section 2.1 was used not in the body but only in the appendix. Thus, it is unnecessary in the body and we moved it from the body to the appendix.
> 2. Although $L^p$ for general $p$ was introduced in the previous manuscript, we used only $L^1$.
> Thus, we defined only $L^1$ space instead of $L^p$.
> 3. We moved some results of Section 2.2 in the previous paper to the appendix (Section A) because they were not used in the body but only in the appendix.
>
> **Comment 1-3**
> >Discussing the invariant case might also help.
>
> We added the explanation about the invariant case in "Inapplicable Cases" of Section 3.3.
>
> **Comment 1-4**
> >I would also suggest rewriting the paper in the following way: start with S,T finite and present all the results. Then discuss the generalization to infinite S,T (which are less useful in my opinion).
>
> As a consequence of careful consideration of the suggestion, we decided not to change the structure but to add Figure 1 in Section 2.2 and many concrete examples in Section 3.3 in order to promote better understanding.
> The following is the reason we did not change the structure of the paper.
>
> First of all, we believe that infinite-dimensional cases (i.e., $\mathcal{S}$ and/or $\mathcal{T}$ are infinite) are as useful as finite-dimensional cases.
> Roughly speaking, finite-dimensional cases can handle only finite-group actions such as permutations of data order.
> On the other hand, infinite-dimensional cases can handle infinite-group actions such as spatial rotation in image data and temporal translation in audio data.
> Moreover, group CNNs with other types of continuous groups have been reported to be useful for multiple applications (e.g., Finzi et al. (2020)).
> Next, we will explain how infinite-dimensional cases naturally appear through the example of 2-dimensional gray-scale image processing in more detail.
> Conventional CNNs for image data have achieved impressive performance, and one of the most significant factors of its success is the fact that the CNNs are translation-equivariant.
> To describe the translation equivariance,
> we need to define the group action of the translation group on pixel coordinates.
> Then, such a situation inevitably becomes an infinite-dimensional case as follows:
> Let $\mathcal{S}\subset \mathbb{R}^2$ be the set of  2-dimensional pixel coordinates of images.
> Then, gray-scale images are represented by elements in $[0,1]^{\mathcal{S}}$, where $[0,1]$ represents the set of gray-scale pixel values.
> If $\mathcal{S}$ is assumed to be finite,
> the set $g+\mathcal{S}:=\{g+s|s\in \mathcal{S}\}$ after translation with an element $0\ne g\in G$ is not included in $\mathcal{S}$, and thus, the action of the translation group $G$ on $\mathcal{S}$ is not well-defined.
> Therefore, $\mathcal{S}$ has to be an infinite set as long as the translation group acts on $\mathcal{S}$.
>
>
> **Comment 1-5**
> >General comment: Please add examples
>
> We added concrete examples in Section 3.3.
>
> **Comment 1-6**
> >Relation to prior work is not discussed. What is the relation of theorem 13 to the results obtained by Maron, Keriven and Ravanbakhsh? Specifically, these works show that in some cases high order tensors are needed for universal approximation by invariant neural networks - can this result be recovered from your results?
>
> As discussed in Section 3.3, our model is a generalization of DeepSets. However, it is not a generalization of the higher-order tensor model. This is due to the fact that in the discussion in Section 3.3, $[n]$ could be expressed in the form of $G/H$. In fact, in the case of a second-order tensor representation of a permutation representation where n is prime, the representation space is $\mathbb{R}^{n^2}$, but it is provable that there is no $H$ where the order of $G/H$ is $n^2$. This phenomenon corresponds to the fact that in some cases a higher-order tensor model is needed. It is future work to do a general involving higher-order tensor models.
>
> **Comment 1-7**
> >The results in this paper seem strong and important, but in its current form it cannot be accepted. I recommend a significant revision of the paper to make it more readable.
>
> We made a significant revision as explained in the replies to each reviewer and the list of changes.

---

### Official Review · AnonReviewer4 · 2020-10-28
**Nice formalism, connection to previous models and some details not clear**

**Rating:** 5
**Confidence:** 3

**Review:**

This paper considers a certain generalization of convolutional neural networks and equivariant linear networks to the infinite dimensional case, while covering also the discrete case, and offers a universality result. In more detail, the paper first characterizes  equivariant maps as the unique  extensions of "generator", namely regular maps that provide target functions (or vector) defined over a basic domain. In other words, any map that takes functions (or vectors) into functions (or vectors) defined over the representatives from the symmetry's equivalent classes can be extended uniquely to an equivariant map by enlarging its target domain according to the equivariance rule. Second, infinite dimensional fully connected networks (FNNs) and general (equivariant) convolution neural networks (CNNs) are described. The main result of the paper is the Conversion Theorem (Theorem 11), and its consequences. The theorem specify the conditions under which an FNN can be approximated by a CNN. Since FNNs are known to be universal this implies universality of CNNs.

I think the general CNN formulation and the Conversion theorem are of merit but i think the paper should undergo a rather serious revision before ready for publication. The main issues (that are detailed below) are: the paper does not sufficiently relate the discussed model or the universality results to previous or concrete models (set and graph NN, equivariant group NN, other unused but potentially useful variations), it does not provide sufficient explanation and justification to the different conditions in Theorem 11, there are some details in the proof and the description of Theorem 11 which are missing/unclear, Theorem 16 has some unclarity. I think attending these will provide a much more accessible and useful paper for the community.

DETAILS.

> Relations to previous work. The paper discuss a rather broad generalization of equivariant network (equations 4 and 5). I think exploring the relations to existing generalization to convolution neural networks and equivariant networks is in order. For example, when taking $\mathcal{S}=\mathcal{T}=[n]$ ($[n]=\{1,2,...,n\}$) and $G=S_n$ (the permutations of $[n]$) equation (4) seems to boil  down to  Deepsets of Zaheer et al. mapping $\mathbb{R}^n\rightarrow \mathbb{R}^n$ equivariantly. Does theorem 1 implies that Deepsets (as equivariant model) is universal (as was proved in several previous works)? In this case, Theorem 3 states that any function $f_1(x)$ can be extended to be equivariant via $f_j(x)=f_1(g\cdot x)$, where $g$ is a permutation such that $g(j)=1$. If this is all required for universality proof this would be a great simplification over previous proofs. This example could help the reader grasp this extension and also relate to previous works on equivariant learning.  What can be said about the equivariant tensor models and graph neural networks using the universality result in this paper? What can be said about the negative results of universality of second order (or higher order) tensor graph networks? Does the method in this paper imply group convolution networks are universal?  Are there any instantiations of the CNN discussed in this paper that are useful although not discussed in previous work? In summary, how the results in this paper relates to known and unknown results of CNN generalizations and what universality proof does it generalize?

> Theorem 11. First, looking at the proof, I feel there is a condition of the FNN $\phi$ that is missing from the theorem's formulation. That is, that the different layers of $\phi$ should be mapping to and from functions on base domains of $G$, $\mathcal{C}(\mathcal{B})$. Without it I can't see how the extension to equivariant maps work. How is it guaranteed that $\phi$ can be extended if internal states are defined on non-base domains? If this is true then how the FNN found from the universality result of FNN (e.g., Theorem 12 and 14) are guaranteed to satisfy this condition? Related to that, I couldn't exactly understand the claim of the second to last layers in the proof of Theorem 11: the first layer outputs a function in $\mathcal{C}(G/H_\mathcal{T}\times \mathcal{B})$, but the second layer of the FNN maps functions from some different domain $\mathcal{C}(\mathcal{B}_2)$. If I understand correctly, the idea of the proof is basically using the extension mechanism of Theorem 3 to extend mapping to functions over base domains to equivariant mappings. This seems to produce a sort of lift in the domain of the target function at every step. Is that correct? Can you provide the proof for a simple example of equivariant networks such as Deepsets? Minor: there are some wrongs equation references in the proof of Theorem 11.

> The condition (C1) in Theorem is not clear to this reviewer. I think some examples and explanation should be provided. I could not really figure out, despite the following paragraphs, in what situations can we verify this condition and in what sense it limits the scope of the theorem. Is there a simple intuition or a way to check such subgroups exists? Are these normal subgroups?  Should $H_\mathcal{T}$ be a strict subgroup of $F_{\mathcal{S}}$? Is this condition holds in standard cases such as $[n]$ and $G=S_n$, or images and $90$ degrees rotations? Does it mean $\mathcal{S}$ has to have a group structure?


> In Theorem 16: How  do we make sure the first layer can be seen as a generator? Can we given $[N]$ the required base structure? Is $[N]$ a base domain of $G$ necessarily? This also relates to the question I asked above about the condition in Theorem 11. Anyway, this proof should be provided in the paper or supplamentary.

> The authors mention that the invariant case cannot be handled. However, invariant functions can be made equivariant by considering the trivial representation, e.g., in the discrete case let $f(x)$ be invariant to $S_n$, then $f_i(x)= f(x)$ for all $i\in [n]$ is equivariant I believe. Can we approximate this equivariant function?

---

> ### Author Response · Authors · 2020-11-22
> **To Reviewer 4**
>
> **Comment 4-1**
> > When taking $\mathcal{S}=\mathcal{T}=[n] ([n]=1,2, \ldots, n)$ and $G=S_{n}$ (the permutations of $\left.[n]\right)$ equation (5) seems to boil down to DeepSets of Zaheer et al. mapping $\mathbb{R}^{n} \rightarrow \mathbb{R}^{n}$ equivariantly.
> Does theorem 1 implies that DeepSets (as equivariant model) is universal?
>
> As reviewer 4 wrote, we can obtain the universality of DeepSets as an application of our result.
> Since we added the explanation about the universality of DeepSets in Section 4.1,
> please see the section for more detail.
>
> **Comment 4-2**
> > What can be said about the equivariant tensor models and graph neural networks using the universality result in this paper?
> What can be said about the negative results of universality of second order (or higher order) tensor graph networks?
> Does the method in this paper imply group convolution networks are universal?
>
> As we explain in Section 3.3, our model can handle the convolution on permutation groups, and we can also prove the universality of the permutation equivariant model.
> However, this is not inconsistent with the negative result since, for example, the second-order tensor action of the permutation action cannot be handled in this formulation as it is.
> This is because the general finite group does not fit this formulation as explained in the paragraph of "Inapplicable cases" of Section 3.3.
> Although this is not the focus of this paper,
> it is possible to extend this formulation to deal with tensor representations of permutation actions and so on in future work.
> This formulation includes graph neural networks.
>
> **Comment 4-3**
> >Are there any instantiations of the CNN discussed in this paper that are useful although not discussed in previous work?
>
> For example, when a group $G$ is a $d$-dimensional Euclidean group $\mathrm{E}(d)$ or a special Euclidean group $\mathrm{SE}(d)$ for $d\in\mathbb{N}$,
> our theory shows that group CNNs are universal approximators of $G$-equivariant maps.
> Although Yarotsky (2018) showed that group CNNs can approximate SE(2)-equivariant maps,
> this result for $d\ge 3$ was not shown in existing studies.
> Since Euclidean groups are often used to represent 3D motion and point cloud,
> group CNNs for Euclidean groups are expected to be useful.
> As another example, a subgroup of the Lorentz group $\mathrm{O}(d,1)$ can be handled as explained in Section 3.3.
> The group acts on the $d+1$-dimensional upper half-plane, which is shown to be suitable for word representations in NLP (Nickel and Kiela (2017)). Since the group action preserves the distance on the upper half-plane, group convolution with the subgroup of the Lorentz group may be useful for NLP.
> Furthermore, since our formulation is quite general,
> it is expected to include other novel and useful CNNs.
>
> **Comment 4-4**
> >In summary, how the results in this paper relates to known and unknown results of CNN generalizations and what universality proof does it generalize?
>
> In existing studies,
> the universality for non-linear equivariant maps has been proved for finite groups, the $d$-dimensional translation group for $d\in\mathbb{N}$, and the $2$-dimensional Euclidean group as explained in Section 1.1.
> Moreover, the universality for the above groups was proven by different methods.
> On the other hand, our model can handle some of the above groups as special cases (as explained in Section 3.3),
> and we can provide a unified proof of the universality of our model for various groups.

---

> ### Author Response · Authors · 2020-11-22
> **To Reviewer 4 (Cont'd)**
>
> **Comment 4-5**
> >The conversion theorem.
> I feel there is a condition of the FNN $\phi$ that is missing from the theorem's formulation.
> That is, that the different layers of $\phi$ should be mapping to and from functions on base domains of $G$, $\mathcal{C}(\mathcal{B}).$
> Without it $\mathrm{I}$ can't see how the extension to equivariant maps work.
> How is it guaranteed that $\phi$ can be extended if internal states are defined on non-base domains?
> If this is true then how the FNN found from the universality result of FNN are guaranteed to satisfy this condition?
> Related to that, I couldn't exactly understand the claim of the second to last layers in the proof of Theorem 9: the first layer outputs a function in $\mathcal{C}\left(G / H\_{\mathcal{T}} \times \mathcal{B}\right)$, but the second layer of the FNN maps functions from some different domain $\mathcal{C}\left(\mathcal{B}\_{2}\right)$.
>
> We carefully checked the proof of Theorem 9 and could not find essential errors.
> However, we noticed that our description and notation may be misleading.
> In the following, we will explain the (possibly) misleading points of the proof.
> Theorem 9 assumes that a group $G$ acts on two sets $\mathcal{S}$ and $\mathcal{T}$, and
> the condition (C1) guarantees the existence of base spaces $\mathcal{B}\_{0}
> := \mathcal{B}\_{\mathcal{S}}$
> and $\mathcal{B}\_L:=\mathcal{B}\_{\mathcal{T}}$.
> Theorem 9 also assumes the existence of an FNN that satisfies (C2).
> In other words,
> Theorem 9 also assumes that
> there exist topological spaces $\mathcal{B}\_{\ell}$ for $\ell=1,\ldots,L-1$
> and affine maps $A\_{1}: E\to \mathcal{C}\_{0}(\mathcal{B}\_{1})$,
> $A\_{\ell}: \mathcal{C}\_{0}(\mathcal{B}\_{\ell-1})\to \mathcal{C\_{0}}(\mathcal{B\_{\ell}})$ for
> $\ell=1,\ldots,L-1$,
> and $A\_L: \mathcal{C}\_{0}(\mathcal{B}\_{L-1})\to \mathcal{C}\_{0}(\mathcal{B}\_{\mathcal{T}})$
> such that the FNN $\phi=A\_{L} \circ \rho \circ A\_{L-1} \circ \cdots \circ \rho \circ A\_{1}:E\to \mathcal{C}\_{0}(\mathcal{B}\_{\mathcal{T}})$ satisfies (C2), where $E\subset \mathcal{C}\_{0}(\mathcal{S})$ is a compact set.
> We note that the sets $\mathcal{B}\_{\ell}$ for $\ell=1,\ldots,L-1$ does not relate to any group action at this point (while $\mathcal{B}\_{\mathcal{S}}$ and $\mathcal{B}\_{\mathcal{T}}$ are defined via the action of a group $G$).
> When we define as $\mathcal{S}\_{\ell}:=G/H\_{\mathcal{T}}\times \mathcal{B}\_{\ell}$ for $\ell=1,\ldots,L-1$,
> the action of $G$ on $\mathcal{S}\_{\ell}$ is naturally defined (by the action of $G$ on $G/H\_{\mathcal{T}}$) and
> the sets $\mathcal{B}\_{\ell}$ become the base space.
> Then,
> convolutions $C\_{\ell}:\mathcal{C}\_{0}(\mathcal{S}\_{\ell-1}) \to \mathcal{C}\_0(\mathcal{S}\_{\ell})$ exist such that the diagram in Figure 2 is approximately commutative in the sense of (20), (21) and (22).
> Here, we emphasize that $\mathcal{B}\_{\mathcal{S}}$ and $\mathcal{B}\_{\mathcal{T}}$ are determined AFTER $\mathcal{S}$ and $\mathcal{T}$ are given
> while $\mathcal{B}\_{\ell}$ is determined BEFORE $\mathcal{S}\_{\ell}$ is defined for $\ell=1,\ldots,L-1$.
> In particular, $\mathcal{S}\_{\ell}$ has a trivial decomposition with the base space $\mathcal{B}\_{\ell}$ by the definition of $\mathcal{S}\_{\ell}$.
>
> We modified the proof of Theorem 9 as above in the revised manuscript.
>
> **Comment 4-6**
> >If I understand correctly, the idea of the proof is basically using the extension mechanism of Theorem 3 to extend mapping to functions over base domains to equivariant mappings.
> This seems to produce a sort of lift in the domain of the target function at every step.
> Is that correct?
>
> Yes, it is correct.
>
> **Comment 4-7**
> >The condition ($C1$) in Theorem is not clear to this reviewer. I think some examples and explanation should be provided. I could not really figure out, despite the following paragraphs, in what situations can we verify this condition and in what sense it limits the scope of the theorem.
>
> We added examples in "Applicable Cases" and the explanation about the limitation of Theorem 9 in "Inapplicable Cases" of Section 3.3.

---

> ### Author Response · Authors · 2020-11-22
> **To Reviewer 4 (Cont'd)**
>
>
> **Comment 4-8**
> >Is there a simple intuition or a way to check such subgroups exists?
> Are these normal subgroups?
> Should $H_{\mathcal{T}}$ be a strict subgroup of $F_{\mathcal{S}}$ ?
> Is this condition holds in standard cases such as $[n]$ and $G=S_{n}$ or images and 90 degrees rotations?
> Does it mean $\mathcal{S}$ has to have a group structure?
>
> Whether the group actions on $\mathcal{S}$ and $\mathcal{T}$ satisfy the condition (C1) depends on the situation and we do not have a general way to check it.
> However, because group theory and representation theory in mathematics covers a vast variety of groups,
> we are expected to have some knowledge about each group action in most cases and then can check the condition (C1).
> The subgroups $H_{\mathcal{S}}$ and $H_{\mathcal{T}}$ are not assumed to be normal.
> We clarified it in footnote 5.
> The condition (C1) is satisfied when the action of $G=S_{n}$ on $[n]$ as explained in Section 3.3.
> The condition (C1) is also satisfied when the action of $C_4$ (90 degrees rotations) on $\mathbb{R}^2$
> because $\mathbb{R}^2 = C_4 \times B_{\mathcal{T}}$, where $B_{\mathcal{T}}$ is the first quadrant.
> In general, $\mathcal{S}$  does not have a group structure but it has a principal bundle structure.
>
> **Comment 4-9**
> >In Theorem 14: How do we make sure the first layer can be seen as a generator? Can we given $[N]$ the required base structure? Is $[N]$ a base domain of $G$ necessarily? This also relates to the question I asked above about the condition in Theorem $9$. Anyway, this proof should be provided in the paper or supplementary.
>
> As we explained in the reply to Comment 4-5, $[N]$ becomes a base space and we modified the proof of Theorem 9.
>
> **Comment 4-10**
> >The authors mention that the invariant case cannot be handled. However, invariant functions can be made equivariant by considering the trivial representation, e.g., in the discrete case let $f(x)$ be invariant to $S\_{n}$, then $f\_{i}(x)=f(x)$ for all $i \in[n]$ is equivariant I believe. Can we approximate this equivariant function?
>
> As reviewer 4 pointed out,
> invariant functions can be made equivariant by considering the trivial representation of $G$ on $\mathcal{T}$ (i.e., $H\_{\mathcal{T}}=G$).
> Then, if (C1) holds, $\mathcal{S}$ also has the trivial representation since $H_{\mathcal{T}} \subset H_{\mathcal{S}}\subset G$.
>
> Moreover, we can show that the condition that $H_{\mathcal{T}} \subset H_{\mathcal{S}}$ is necessary for the conversion theorem.
> To prove this, we will construct a counterexample for the conversion theorem if $H_{\mathcal{T}} \subset H_{\mathcal{S}}$ does not hold.
> We consider one of the simplest invariant settings.
> Let $\mathcal{S}=\mathcal{T}=G=\{\pm 1\}$.
> Suppose that the action of $G$ on $\mathcal{S}$ is multiplication and that on $\mathcal{T}$ is trivial.
> Then,
> $\mathcal{S}=G/H\_{\mathcal{S}}\times \mathcal{B}\_{\mathcal{S}}$ and $\mathcal{T}=G/H\_{\mathcal{T}}\times \mathcal{B}\_{\mathcal{T}}$
> hold for $H\_{\mathcal{S}}=\{1\}$, $\mathcal{B}\_{\mathcal{S}}=\{\ast\}$, $H\_{\mathcal{T}}=G=\{\pm 1\}$, and $\mathcal{B}\_{\mathcal{T}}=\mathcal{T}=\{\pm 1\}$, where $\{\ast\}$ is a singleton.
> However, since $H\_{\mathcal{T}} \nsubseteq H\_{\mathcal{S}}$, (C1) does not hold.
> Here, we note that $\mathcal{C}\_{0}(\mathcal{S})=\mathcal{C}\_{0}(\mathcal{T})=\mathcal{C}\_{0}(\mathcal{B}\_{\mathcal{T}})=\mathbb{R}^2$ and $R\_{\mathcal{B}\_{\mathcal{T}}}=\mathrm{id}$.
> Furthermore, we note that (C2) is always satisfied because $\mathcal{S}$ is finite.
> Then,
> if the consequences of Theorem 9 hold,
> for an arbitrary $\epsilon>0$, an arbitrary  compact set $E\subset \mathcal{C}\_{0}(\mathcal{S})=\mathbb{R}^2$, and an arbitrary FNN $\phi:E\to \mathcal{C}\_{0}(\mathcal{T})=\mathbb{R}^2$ with Lipschitz activation function,
> there exists a CNN $\Phi:E\to \mathcal{C}\_{0}=\mathcal{T}$ such that the inequality $\|\Phi - \phi\|\_{\infty}\le \epsilon$ in (8) holds.
> However, the inequality does not hold for a non-invariant FNN $\phi$ and small $\epsilon>0$ because the CNN $\Phi$ is in the class of invariant functions.
>
> Thus, the condition (C1) is essential in Theorem 9, and the invariant case is excluded due to (C1).
> We added the explanation about the invariant case in "Inapplicable Cases" of Section 3.3.

---

### Author Response · Authors · 2020-11-22
**To All Reviewers**


We thank all reviewers for the positive feedback and helpful comments.
We have revised our paper by taking into account the suggestions.
The list of major changes are summarized as follows.

- Section 1.1: We modified the description of every paragraph about related works.

- Section 1.2:  We added the organization of the paper and summarized our contribution.

- Section 2.2: We added Figure 1 to visualize equivariant maps between infinite-dimensional function spaces.
We simplified Theorem 3 and moved some results in the previous manuscript to Appendix (Section A.1).

- Section 3.1: We added the explanation of affine maps in finite dimension as the equation (3).

- Section 3.3: We refined the statement of Theorem 9 (conversion theorem) without mathematically changing the assumptions and the consequences.
We added examples of concrete groups that satisfy the conditions in Theorem 9.
We added the explanation of inapplicable cases about Theorem 9. In particular, we added the description of invariant maps.

- Section 4.1: We derived the universality of DeepSets as a corollary of our results.

- Section 4.2: We added the explanation about the novelty of our results and concrete examples of group CNNs that were not discussed in previous work.

- Section 5: We modified the latter part of the section.

- Section B: We added Figure 2 to visualize the relation between FNNs and CNNs in the conversion theorem (Theorem 9).

- Section C: We provided proof of the universality of DeepSets as a corollary of our results.

In addition, we corrected typos throughout the paper.

---

### Decision · Program_Chairs · 2021-01-07
**Final Decision**

**Decision:**

Reject

**Comment:**

This work studies the question of universal approximation with neural networks under general symmetries. For this purpose, the authors first leverage existing universal approximation results with shallow fully connected networks defined on infinite-dimensional input spaces, that are then upgraded to provide Universal Approximation of group-equivariant functions using group equivariant  convolutional networks.

Reviewers were all appreciative of the scope of this paper, aimed at unifying different UAT results under the same underlying 'master theorem', bringing a much more general perspective on the problem of learning under symmetry. However, reviewers also expressed concerns about the accessibility and readibility of the current manuscript, pointing at the lack of examples and connections with existing models/results. Authors did a commendable job at adding these examples and incorporating reviewers feedback into a much improved revision.

After taking all the feedback into account, this AC has the uncomfortable job of recommending rejection at this time. Ultimately, the reason is that this AC is convinced that this paper can be made even better by doing an extra revision that helps the reader navigate through the levels of generality. As it turns out, this paper was reviewed by three top senior experts at the interface of ML and groups/invariances, who themselves found that the treatment could be made more accessible --- thus hinting a difficult read for non-experts. In particular, the main result of this work (theorem 9) is based on a rather intuitive idea (that one can leverage UAT for generic neural nets on the generator of an equivariant function), that requires some technical 'care' in order to be fleshed out. The essence of the proof can be conveyed in simple terms, after which following through the proof is much easier. Similarly, the paper quickly adopts an abstract (yet precise) formalism in terms of infinite-dimensional domains, which again clouds the essential ideas in technical details. While the paper now contains several examples, this AC believes the authors can go to the extra mile of connecting them together, and further discussing the shortcomings of the result --- in particular, the remarks on tensor representations and the invariant case are of great importance in practice, and should be discussed more prominently. Finally, while this work is only concerned with universal approximation, an important aspect that is not mentioned here is the quantitative counterpart, ie what are the approximation rates for symmetric functions under the considered models.